# Biomechanical Comparison of Vertebroplasty, Kyphoplasty, Vertebrae Stent for Osteoporotic Vertebral Compression Fractures—A Finite Element Analysis

Jen-Chung Liao [1,*,†], Michael Jian-Wen Chen [1,†], Tung-Yi Lin [2,*] and Weng-Pin Chen [3,*]

1. Department of Orthopedic Surgery, Bone and Joint Research Center, Chang Gung Memorial Hospital, Chang Gung University, Taoyuan 33302, Taiwan; mchen115@gmial.com
2. Department of Orthopedic Surgery, Chang Gung Memorial Hospital (Keelung Branch), Chang Gung University, Taoyuan 33302, Taiwan
3. Department of Mechanical Engineering, National Taipei University of Technology, Taipei 10608, Taiwan
* Correspondence: jcl1265@adm.cgmh.org.tw (J.-C.L.); b9002095@cgmh.org.tw (T.-Y.L.); wpchen@ntut.edu.tw (W.-P.C.)
† Co-first authors.

**Abstract:** Vertebroplasty (VP), balloon kyphoplasty (BKP), and vertebral stent (VS) are usually used for treating osteoporotic compression fractures. However, these procedures may pose risks of secondary adjacent level fractures. This study simulates finite element models of osteoporotic compression fractures treated with VP, BKP, and VS Vertebral resection method was used to simulate vertebra fracture with Young's modulus set at 70 MPa to replicate osteoporosis. A follower load of (1175 N for flexion, and 500 N for all others) was applied in between vertebral bodies to simulate the muscle force. Moment loadings of 7.5 N-m in flexion, extension, lateral bending, axial rotation were applied respectively. The VS model had the highest von Mises stresses on the bone cement under all different loading conditions (flexion/5.91 MPa; extension/3.74 MPa; lateral bending/3.12 MPa; axial rotation/3.54 MPa). The stress distribution and maximum von Mises stresses of the adjacent segments, T11 inferior endplate and L1 superior endplate, showed no significant difference among three surgical models. The postoperative T12 stiffness for VP, BKP, and VS are 2898.48 N/mm, 4123.18 N/mm, and 4690.34 N/mm, respectively. The VS model led to superior surgical vertebra stiffness without significantly increasing the risks of adjacent fracture.

**Keywords:** osteoporosis; compression fracture; vertebroplasty; balloon kyphoplasty; vertebral stent system; adjacent fracture



## 1. Introduction

Osteoporosis is an increasingly prevalent disease globally. When bone becomes thin and brittle due to decreased bone density, it may lead to fractures, especially at the forearm, femur and spine vertebrae. Vertebral compression fractures (VCF) occur due to axial loading forces that result in vertebral body height decrease, usually located at the anterior column with an anterior wedge appearance. Regardless of the degree of vertebral height decrease, VCFs may elicit debilitating symptoms, including back soreness, severe pain, functional limitations, and decreased life quality [1]. Treatment options for vertebral compression fracture (VCF) consist of conservative and surgical treatments. At the acute stage of VCF, bed rest, brace support, and prescription of analgesics are generally recommended, however, if the patients' symptoms persist after conservative treatment for several months, surgical treatment may be needed. Cement augmentation with vertebroplasty (VP), balloon kyphoplasty (BKP), or vertebrae stent (VS) (Spinejack system, Stryker Corp., Kalamazoo, MI) have been widely used procedures to treat patients' pain and improve their quality of life [2–4]. VP is performed after administration of a local anesthetic combined with intravenous narcotic or sedative drugs. The needle entry site over

the pedicle is localized under fluoroscopic guidance, and then polymethylmethacrylate (PMMA) delivery into the vertebrae body is done to stabilize the fracture and alleviate pain. BKP uses a high-pressure balloon to force the collapsed vertebral body back to its original height by creating a cavity that is subsequently filled with the PMMA. In contrast, vertebrae stent (VS) (SpineJack) utilizes a cranio-caudal expandable implant to increase the body height followed by filling the intravertebral stent with PMMA [5].

In the past literature, there has been numerous clinical and biomechanical studies that compare the outcomes for both VP and BKP [6–11]. Though clinical results of both VP and BKP were acceptable, both methods could not prevent the development of adjacent vertebrae fractures. These adjacent fractures are not only due to bone fragility, but also can be caused by the new increased rigidity of the vertebrae filled with bone cement [12,13]. In the past decade, the vertebrae stent has become a novel surgical method for vertebral compression fracture surgery, as it has also been clinically proven to be effective for vertebral compression fracture treatment [14,15]. The design concept of the vertebral stent is to increase the stability through the complex of bone cement and stent, to prevent the deflating effect of balloons, and to provide more correction of local alignment [16]. Theoretically, a vertebral stent makes the vertebral body more evenly stressed, and is expected to reduce the complications at both the fractured vertebrae and the adjacent vertebrae. To our knowledge, comparative studies using finite element analysis for the VP, BKP and VS procedures are scarce and limited. The aim of this study was to simulate a finite element model of an osteoporotic compression fracture at the thoracolumbar region treated with these three surgical methods (VP, BKP, and VS). Stress distribution and stiffness of the treated vertebrae and the adjacent vertebral segments were simulated under compressive loading simulation to evaluate the possibility of developing secondary fractures. The findings of this study may guide future treatment decisions for surgeons to provide the most appropriate intervention.

## 2. Methods

### 2.1. Models

Three models were established to simulate thoracolumbar osteoporotic compression fracture with different cement augmentations surgeries: (1) VP; (2) BKP; (3) VS Radiographs representing these three types of surgical procedures are shown in Figure 1. These three surgical methods are described briefly. VP: a cannulated trocar was inserted into the injured vertebrae, a certain amount of bone cement was injected directly into the injured vertebrae through the trocar. BKP: a cannulated trocar was inserted in to the injured vertebrae, an uninflated balloon was inserted, the collapsed vertebral body was reduced by inflating the balloon, the balloon was deflated and removed, then the bone cement was injected into the space of the injured vertebrae. VS: a cannulated trocar was inserted into the injured vertebrae, the expandable stent was inserted into an ideal position of the fractured vertebrae, the collapsed vertebrae was reduced by expanding the stent, and the bone cement was injected into the vertebrae and the stent.

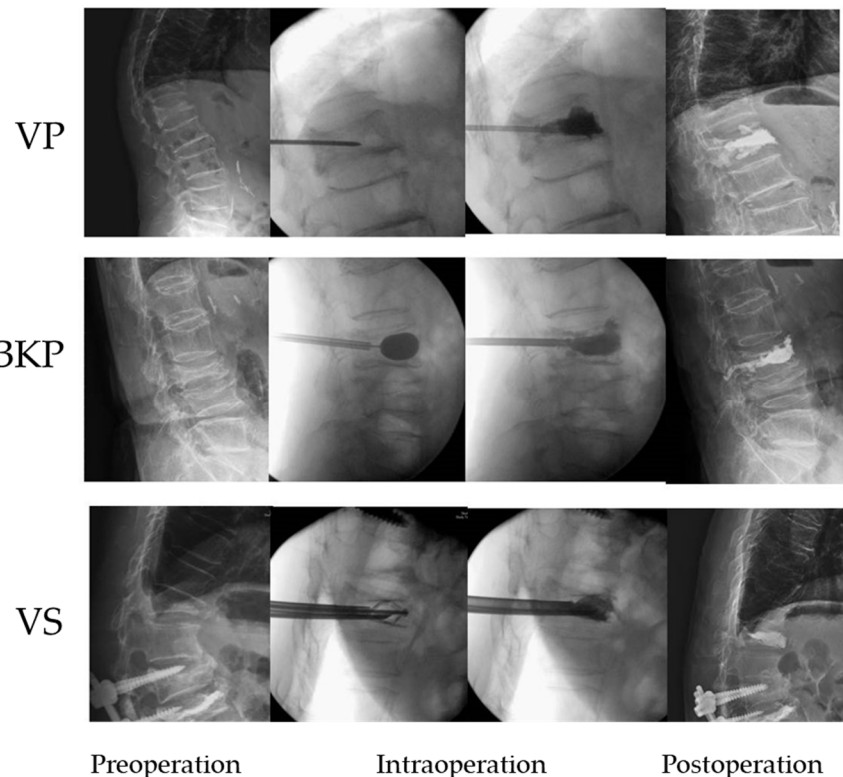

**Figure 1.** Radiographs representing three types of cement augmentation technique. VP: Vertebropalsty, BKP: Balloon kyphoplasty, VS: Vertebrae stent.

2.1.1. Establishment of an intact normal thoracolumbar spine in the FE model

An intact normal spine finite element (FE) model was constructed by 1 mm slice-interval cross-sectional computed tomography (CT) spine images of a 65-year-old male obtained from the Visible Human project of the U.S. National Library of Medicine (NLM, NIH, Bethesda, MD, USA), as shown in Figure 2. The process of creating a finite element model of T10 to L2 were briefly described as following. (1) CT scans were imported into Amira software (Visage Imaging, Carlsbad, CA, USA). Vertebral boundaries of interest were identified with multiple images to form a three-node triangular surface model. (2) The surface model was imported into SolidWorks (SOLIDWORKS Corp., Boston, MA, USA), to further create a three-dimensional solid model of the thoracolumbar spine. (3) The solid model of the thoracolumbar spine was imported into HyperWorks 10.0 (Altair Engineering, Inc., Troy, MI, USA) to form the FE model. CT images do not provide structural contours of the intervertebral discs. Therefore, the geometric characteristics of intervertebral discs were created according to data from Chosa et al. [17]. The volume ratio of the annulus fibrosus to the nucleus pulposus was set 6:4, the thickness of the cortical bone was set to 1 mm, and the endplate was set to 0.5 mm, respectively. (4) Finally, FE model was imported into Abaqus FE analysis software (Abaqus/CAE v.6.10; Simulia Corp., Providence, RI, USA) for further analysis.

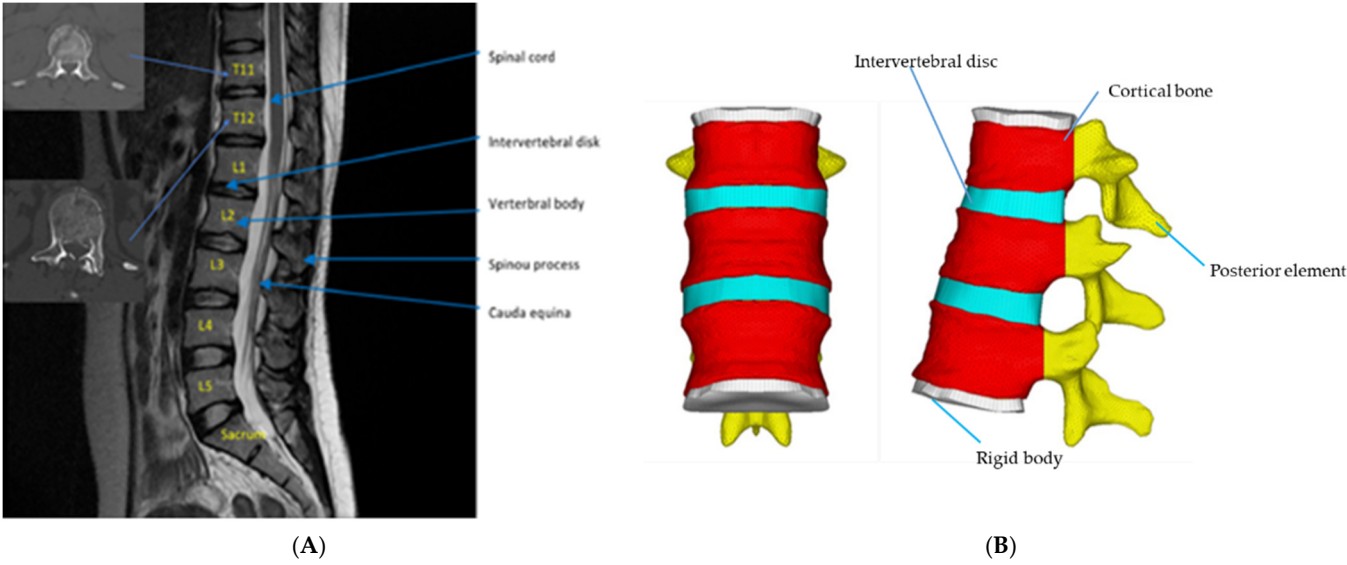

**Figure 2.** (**A**) CT image of the human vertebrae (**B**) T11-L1 Finite element model.

### 2.1.2. Establishment of T12 Injured Vertebra FE Model

Vertebral resection method was used to simulate vertebra fracture in this study. One-half of the sponge bone of the T12 vertebra was removed to weaken the vertebral strength. Structure of the posterior part was reserved to simulate thoracolumbar compression fracture.

### 2.2. Material Parameter Setting

The mechanical properties for each part of the FE model were assigned according to the study by Cho et al. [18] except the cortical and cancellous bone portions were assigned to be the same low value of 34 MPa in order to simulate severe osteoporosis, as suggested by Rohlmann et al. [19]. The Young's moduli and Poisson's ratios of end-plate, intervertebral disc, posterior bone element, cortical bone, and cancellous bone were set respectively. These models are assumed to be linear elastic, isotropic, and homogeneous. Table 1 summarizes the material parameters of this study.

**Table 1.** Material Properties of the Finite Element Model [18,19].

| Component | Young's Modulus (MPa) | Poisson Ratio ($\upsilon$) | Element Type |
|---|---|---|---|
| Cortical bone | 34 | 0.3 | Tetrahedral |
| Cancellous bone | 34 | 0.3 | Tetrahedral |
| Posterior element | 2345 | 0.25 | Tetrahedral |
| Endplate | 670 | 0.3 | Tetrahedral |
| Annulus substance | 5 | 0.45 | Hexahedral |
| Nucleus | 9 | 0.4 | Hexahedral |
| Annulus fiber | 455 | 0.3 | Surface |
| Bone cement | 3000 | 0.41 | Tetrahedral |
| Vertebrae stent (SpineJack) | 113,800 | 0.34 | Tetrahedral |
| Rigid body | 10 | 0.4 | Pentahedron |

### 2.2.1. Establishment of VP Model in T12 Compression Fracture

For compression fractures, the VP procedure uses PMMA bone cement as a filler, finds the injectable point under fluoroscope guide, and injects the bone cement with a percutaneous needle to restore the original height of the vertebral body. Kim et al. proposed that under the same compressive load, the rigidity of the treated vertebral segment will be closest to the normal uncompressed vertebra when the bone cement accounts for 30% of the original spongy bone space [20]. In this study, 30% of the cancellous bone space in the T12 vertebral body was removed, and PMMA bone cement (Young's modulus: 3000 MPa, Poisson's ratio: 0.41) was added to simulate the finite element model of osteoporosis after

vertebroplasty; bone cement is directly contacted with the spongy bone in the vertebral body and then solidified. The element type of bone cement is set in the same way as the spongy bone element.

### 2.2.2. Establishment of BKP Model in T12 Compression Fracture

The difference of BKP from VP is that before the bone cement is delivered, the collapsed vertebral body is expanded to the expected height with a special balloon, and then the bone cement is injected into the opened cavity. In this study, 50% volume in the cancellous bone space of T12 vertebral body was excavated in the shape of sphere; then bone cement elements conforming to the spherical shape were created and put into the cavity to simulate bone cement injection, and bone cement with 40% volume of the injured vertebra was used to augment the T12 body, which is similar to that proposed by Purcell et al. [21].

### 2.2.3. Establishment of VS Model with Bone Cement Augmentation in T12 Compression Fracture

In this study VS refers to the use of a commercialized implantable titanium vertebral augmentation device (using the SpineJack system) for treating VCF. The geometry of the vertebral stent SpineJack (VS) Model was created by scanning the SpineJack implant by using a 3D reverse engineering scanning device (ATOS, GOM Inc., Braunschweig, Germany) reverse and the Solidworks CAD software. In this study, 50% of the volume in the cancellous bone space of T12 vertebral body was excavated; SpineJack was assumed to restore 45% of the damaged T12 vertebra and bone cement was added to strengthen it. Figure 3 demonstrated finite element models of these three models.

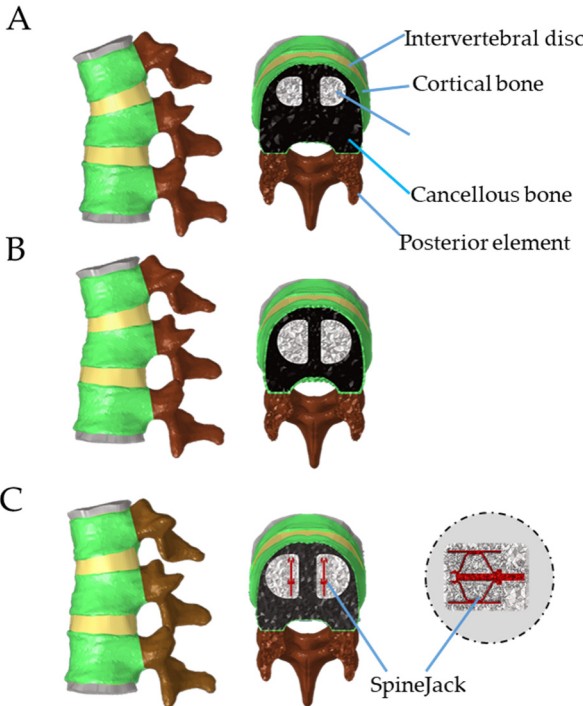

**Figure 3.** T11-L1 finite element model representing three types of cement augmentation technique. (**A**): Vertebroplasty (VP). (**B**): Balloon kyphoplasty (BKP). (**C**): Vertebrae stent (VS).

The element and node numbers for the final FE models used were ranging from 339,429–546,351 for element numbers and 83,472–114,961 for node numbers. Final element densities size of 1.5 mm for vertebral bodies, 1 mm for bone cement, 1.5 mm for endplate, and 0.3 mm for SpineJack were used respectively for different FE models.

### 2.3. Loading and Boundary Condition Settings

The loading conditions are set as follows: (1) A follower load of (1175 N for flexion, and 500 N for all others) was applied in between vertebral bodies to simulate the muscle force. The nodes on the bottom surface of the L1 vertebra were fully constrained as boundary condition. (2) Moment loadings of 7.5 N-m in flexion, extension, lateral bending, axial rotation were applied respectively according to the literature by Rohlmann et al. [19]. The stress distributions on the implant-cancellous bone interfaces were compared under dif-ferent loading conditions, so as to assess the location of the fracture that may occur again in the treated segment. Under different loading conditions, the stress distributions on the adjacent segments (T11 and L1) after implantation were compared for different cement augmentation treatments. It was aimed to observe whether there is a "load shift" phe-nomenon that may cause high stress site to transfer to the internal vertebral body of adja-cent segments, thereby posing a risk of secondary fracture in the adjacent segments.

## 3. Results

The biomechanical performance of the surgical segment and adjacent vertebral bodies were investigated, including von Mises stress distribution and compressive stiffness, in order to understand the effects of different loading conditions and to evaluate the risk of secondary fracture.

### 3.1. Maximum Bone Cement Stress

From Figure 4A–D, the maximum stresses in the bone under flexion loading for each surgery model are: 3.9 MPa for the VP model, 3.59 MPa for the BKP model, and 5.91 Mpa for the VS model; when under extension loading for the VP model it is 2.37 MPa, 3.61 MPa for the BKP model, and 3.74 MPa for the VS model; when under lateral bending conditions for the VP model it is 2.48 MPa, 2.73 MPa for the BKP model, and 3.12 MPa for the VS model; when under axial rotation conditions for the VP model it is 2.13 MPa, 1.85 MPa for the BKP model, and 3.54 MPa for the VS model.

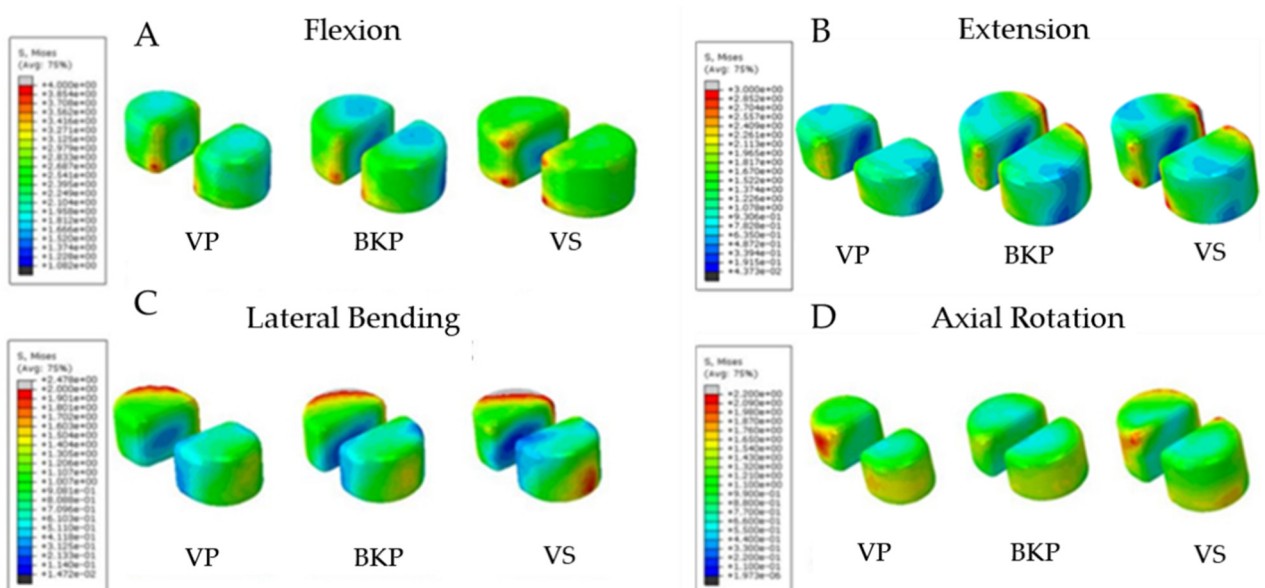

**Figure 4.** Finite element stress nephogram of bone cement. (**A**): Flexion. (**B**): Extension. (**C**): Lateral bending. (**D**): Axial rotation.

### 3.2. Maximum Adjacent Segment Interfacial Stress Distribution

The interfacial stress distributions at the adjacent lower T11 and upper L1 endplates were investigated to evaluate the risk of adjacent level fracture.

### 3.2.1. Maximum Interfacial Stress at Lower T11 Endplate

From Figure 5A–D, the maximum interfacial stress at the Lower T11 endplate under flexion loading for the VP model is 19.52 MPa, 18.46 MPa for the BKP model, and 20.97 MPa for the VS model; under extension conditions for the VP model it is 13.41 MPa, 13.99 MPa for the BKP model, and 12.25 MPa for the VS model; under lateral bending conditions for VP it is 11.61 MPa, 10.91 MPa for the BKP model, and 12.19 MPa for the VS model; under axial rotation conditions for the VP model itis 8.74 MPa, 9.13 MPa for the BKP model, and 8.63 MPa for the VS model.

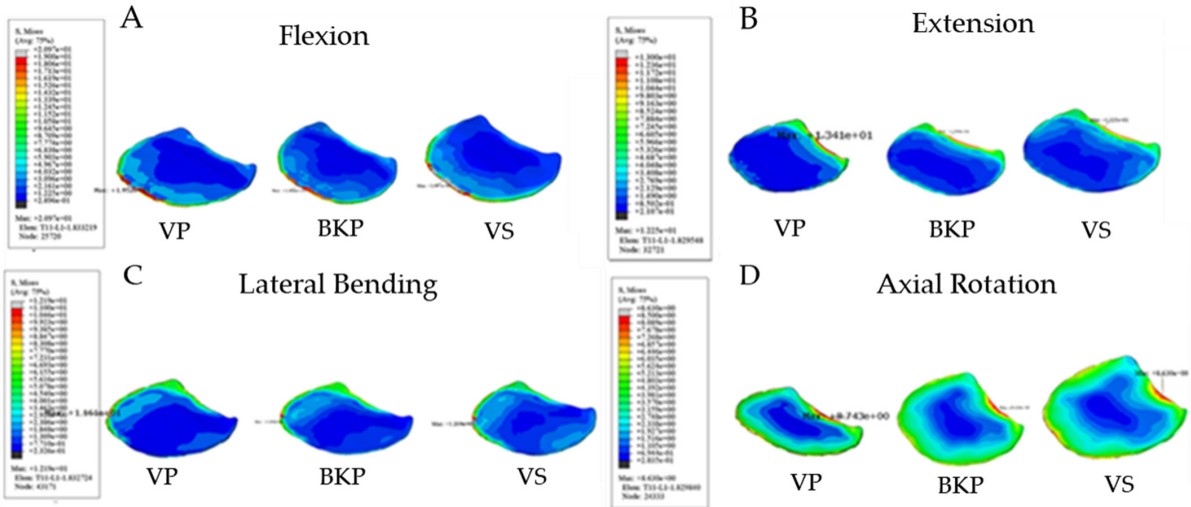

**Figure 5.** Finite element stress nephogram of interfacial stress at lower T11 endplate. (**A**): Flexion. (**B**): Extension. (**C**): Lateral bending. (**D**): Axial rotation.

### 3.2.2. Maximum Interfacial Stress at Upper L1 Endplate

From Figure 6A–D, the maximum upper L1 endplate interfacial stress under flexion conditions for the VP model is 13.74 MPa, 13.79 MPa for the BKP model, and 13.5 MPa for the VS model; under extension conditions for the VP model it is 13.41 MPa, 13.52 MPa for the BKP model, and 14.27 for the VS model; under lateral bending conditions for the VP model it is 11.34 MPa, 11.36 MPa for the BKP model, and 11.17 MPa for the VS model; under axial rotation conditions for the VP model it is 8.1 MPa, 9.13 MPa for the BKP model, and 8.63 MPa for the VS model.

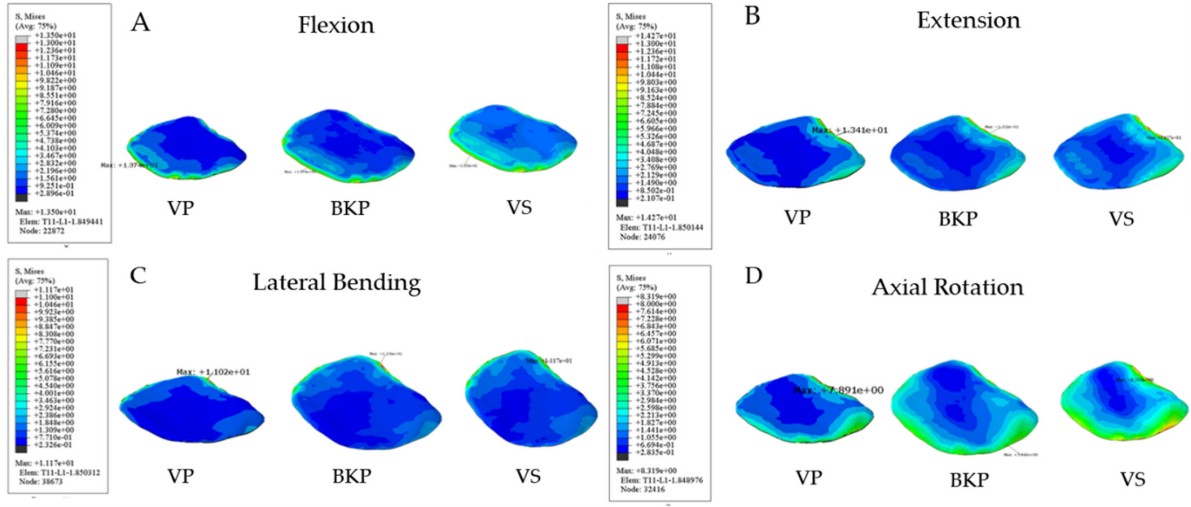

**Figure 6.** Finite element stress nephogram of interfacial stress at upper L1 endplate. (**A**): Flexion. (**B**): Extension. (**C**): Lateral bending. (**D**): Axial rotation.

### 3.2.3. Surgical Segment Vertebra Stiffness

The postoperative T12 stiffnesses (slopes of the force-displacement curves) of the three different surgical interventions are shown in Figure 7. The stiffness values could help understanding the biomechanical behavior of the injured vertebra under 1000 N compressive force. The stiffness for the VP, BKP, and VS groups are: 2898.48 N/mm, 4123.18 N/mm, and 4690.34 N/mm, respectively.

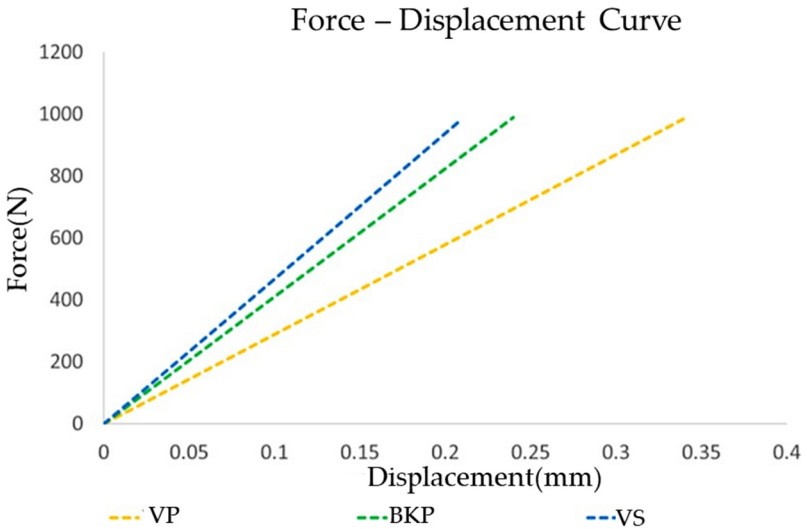

**Figure 7.** Force-displacement curve of the treated vertebrae after three different surgical procedures.

Table 2 summarizes the maximal stress on the T12 cement surface, T11 lower endplate and L1 upper endplate in these three models.

**Table 2.** Maximum von Mises stresses of the three surgical models (MPa).

|  | Flexion | Extension | Lateral Bending | Axial Rotation |
|---|---|---|---|---|
| Bone cement |  |  |  |  |
| VP | 3.90 | 2.37 | 2.48 | 2.13 |
| BKP | 3.59 | 3.61 | 2.73 | 1.85 |
| VS | 5.91 | 3.74 | 3.12 | 3.54 |
| Lower T11 endplate |  |  |  |  |
| VP | 19.52 | 13.41 | 11.61 | 8.74 |
| BKP | 18.46 | 13.99 | 10.91 | 9.13 |
| VS | 20.97 | 12.25 | 12.19 | 8.63 |
| Upper L1 endplate |  |  |  |  |
| VP | 13.74 | 13.41 | 11.34 | 8.1 |
| BKP | 13.79 | 13.52 | 11.36 | 5.84 |
| VS | 13.50 | 14.27 | 11.17 | 8.32 |

VP: Vertebroplasty; BKP: Balloon kyphoplasty; VS: Vertebrae stent.

## 4. Discussion

VP and BKP are common interventions for the treatment of osteoporotic compression fracture of the spine vertebrae. VS can achieve controlled anatomical restoration before bone cement augmentation and has gained popularity recently. Although the height restoration and augment material property may vary, these interventions all use bone cement to stabilize fracture cracks. The aim of this study was to investigate the biomechanical effects on osteoporotic compression fracture of these different interventions.

Clinical studies of different cement augmentation procedures have been encouraging. Several studies have suggested both VP and BKP not only improved quality of life, pain

relief, improved functionality, and restored vertebral body height [22]. A meta-analysis by Zhao et al. showed patients treated with BKP are more effective according to the long term Visual Analogue Scale (VAS), Oswestry Disability Index (ODI), improved kyphosis angle, mean vertebral body height, and experience significantly reduced risk of cement leakage [23]. Clinical outcomes of VKP and VS (SpineJack) cement augmentations were also compared in previous studies. Noriega et al. demonstrated both procedures were safe and led to significant clinical improvement for patients with osteoporotic vertebral compression fractures [5]. In 2019, a prospective, international, randomized study compared an implantable titanium vertebral augmentation device versus BKP in the reduction of vertebral compression fractures and found non-inferiority of the titanium augmentation device with an excellent risk/benefit profile for results up to 12 months [24]. Furthermore, in vitro biomechanical cadaver studies demonstrated height restoration was significantly better in the VS (SpineJack) group compared with the BKP group [25]. The clinical implications include a better restoration of the sagittal balance of the spine and a reduction of the kyphotic deformity. From our results, the VS group has the highest von Mises stress among the three surgical interventions of 5.94 MPa. The usual maximum compressive strength of PMMA is 93 MPa, which is far more than the measured maximum stress in the VS group [26]. Therefore, it is reasonable that the compressive strength of PMMA is well within the safe range for all three surgical procedures.

The effect of cement augmentation on the adjacent vertebral body has been debated. Many studies have reported the incidence of adjacent level fractures, but the results were not consistent. Ma et al. conducted a meta-analysis which encompassed 12 studies with 1081 patients [27]. They concluded that BKP and VP are both safe and effective procedures for treating osteoporotic vertebral compression fractures. There were no statistical differences in VAS, ODI, cement leakage rates, and adjacent vertebral fracture rates. Similarly, Zhao et al. found both VP and BKP had a similar effect on short-term pain relief, posterior vertebral body weight, and adjacent-level fractures [23]. In this study, the stress distributions of the adjacent lower T11 and upper L1 endplate also showed no significant difference among the surgical intervention groups, implying no significant effect on the risks of adjacent fracture.

This study also showed that compression fractures treated with VS have the highest stiffness of 4690.34 N/mm. According to a study on the stress burden of the spine after VP and BKP by Rohlmann et al., the stiffness and strength of the vertebrae increase as the bone cement volume increases [8]. Their probabilistic and sensitivity study suggested that in cement augmentation procedures, the maximum stresses moderately depend on the injected cement volume. The VS group has the most amount of cement augmentation and VP has the least amount. According to Rotter et al., during intraoperative period, VS (SpineJack) could preserve the maximum height gain significantly better than BKP, which creates cavity absence of load-bearing after balloon deflation and before cement injection [28]. Furthermore, the additional support of titanium alloy implant also contributes to a higher stiffness for the VS group. Interestingly, the results of this biomechanical study also suggest a superior body height maintenance ratio for the VS surgical model due to a higher stiffness.

However, several previous articles have reported the possible risk factors for recollapse of cemented vertebrae after percutaneous cement augmentation. In 2011, Chen et al. retrospectively reviewed 1800 patients after a 2-year follow-up. The incidence of refracture of the same vertebra after VP with an incidence rate of 0.56% [29]. Osteonecrosis, greater anterior vertebral height restoration, lesser kyphosis angle correction and cystic filling pattern were found to be significant risk factors for recollapse. As for BKP, Lavelle et al. reported a 10% incidence recurrent fracture after BKP of a previously operated vertebra primarily within the first 90 days after surgery [30]. The study by Kim et al. demonstrated the presence of intervertebral cleft and non-PMMA-endplate-contact may contribute to future recompression of BKP treated vertebrae [31]. Li et al. conducted a risk factor analysis for re-collapse of cemented vertebrae after percutaneous VP or BKP [32]. Low bone mineral

density, percutaneous BKP, and low volume cement injection were found to be associated with high risk of recollapse. Therefore, cautious interpretation of patients with the above risk factors is imperative to prevent deterioration of fracture conditions.

There are several limitations to this study that need to be considered. The creation of a finite element model requires several simplifications and assumptions, including a homogeneous cement filling model and orthotropic material simulation. Physiological boundary conditions, cortical and trabecular bone region distinction, subsequent bone remodeling and pre-existing damage may be neglected. Furthermore, as of many other biomechanical studies, our finite element analysis was limited to a single thoracic vertebral segment and may not be generalized to other vertebrae. Despite the shortcomings mentioned above, this is the first study comparing the stress distribution and maximum von Mises stresses on the treated and adjacent vertebrae among VP, BKP, and VS using finite element analysis. This biomechanical study can improve the understanding of cement augmentation on vertebral compression fracture. To make the results from this finite element analysis clinically applicable, we encourage future multicenter studies with long term follow up to confirm these findings. The interpretation of this finite element analysis can only explain immediate post-operative results but cannot predict long-term clinical effects. According to current literature review, all three surgical procedures lead to good clinical results. This is in line with the low stress distribution values found in our results. Based on clinical experience, the three procedures differ mostly by the restored vertebral height and the amount of bone cement injection. The bone cement will gradually compress over time and the height will decrease. However, metal stent covered with bone cement provides continued support to maintain vertebral height. This is the reason for superior stiffness of the VS group and may further imply long-term clinical benefits.

## 5. Conclusions

According to this finite element analysis, treating compression fracture with vertebral stent led to superior surgical segment vertebra stiffness when compared to vertebroplasty and balloon kyphoplasty. Adjacent endplate interfacial stress distribution was not significantly different among these surgical interventions.

**Author Contributions:** J.-C.L. and W.-P.C. designed the study. J.-C.L. and M.J.-W.C. wrote the manuscript. T.-Y.L. reviewed and edited the manuscript. All authors have read and agreed to the published version of the manuscript.

**Funding:** This study was supported by Chang Gung Memorial Hospital Research Fund (CR-RPG2H0091).

**Institutional Review Board Statement:** Not applicable.

**Data Availability Statement:** The data used to support the finding of this study are available from the corresponding author upon request.

**Conflicts of Interest:** The authors declare that there no conflicts of interest regarding the publication of this paper. All methods were carried out in accordance with relevant guidelines and regulations. All experimental protocols were approved by a named institutional and/or licensing committee.

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
