# Peer review of "Biomechanical Comparison of Vertebroplasty, Kyphoplasty, Vertebrae Stent for Osteoporotic Vertebral Compression Fractures—A Finite Element Analysis"

_applsci, doi:10.3390/app11135764_

Round 1

Reviewer 1 Report

The manuscript is devoted to the investigation, by means of finite element analysis, of the mechanical performances of three different procedure adopted to cure osteoporotic compression fracture.

The overall quality of the manuscript can be improved, given the fact that some of the methodological choice made by the Authors are not very clear.

The paper is written with an acceptable English language; however some sentences are long and difficult to understand.

Some aspects of the submitted paper are not very clear and must be specified or corrected. Here are reported some considerations:

  1. Abstract: Correct “Mpa” with “MPa”.
  2. Introduction, lines 52-53: While the Authors provided an acceptable view of the medical aspect, no detailed information of the state of art is provided. Please, improve this aspect. The authors may find useful the following paper:
    1. https://doi.org/10.1142/S021951941750107X
    2. Becker, Stephan, et al. "Disadvantages of Balloon Kyphoplasty with PMMA-a Clinical and Biomechanical Statement." J. Miner. Stoffwechs 18 (2011): 9-12.
    3. http://dx.doi.org/10.4236/jbise.2016.910043
  3. Methods, Models: This reviewer suggest to move the detailed description of the different treatments in this section, improving the description of Figure 1.
  4. Figure 1: the “Figure 1” above the figure is redundant, remove it.
  5. Methods, lines 68-89: This reviewer suggest to add a figure to improve the description of the creation of the FE model, from CT scan to mesh.
  6. Table 1: Cortical and cancellous bone have the same mechanical properties which differ from the adopted values by Qiu et al., it is correct?
  7. Page 4, lines 124-125: Please, report the source for the related literature and data.
  8. Figure 2: Please, report the in the figure the different meanings for the coloured areas. Remove “Figure 2”
  9. Section 2.3, “Loading and boundary conditions settings”: It is not clear why the authors adopted these values of forces and moments. Please, report the source and explain the physiological conditions that they simulate.
  10. Figure 3: Remove “Figure 3” and add the letters. There is a repetition of “Lateral Bending”. Adopt VS instead of Spinejack.
  11. Figures 4 and 5: Remove “Figure x” and add the letters. The images are very small and with a poor quality. Please, improve it. Adopt VS instead of Spinejack.
  12. Figure 6: Remove “Figure 6”.
  13. Results: No observations were made on the possible crack initiation site. Please, improve this aspect.
  14. Page 8, line 210: Please, specify the meanings of ODI.

For all the previous reasons, the reviewer recommends Major amendments of paper for publication in Applied Sciences.

Author Response

We would like to thank the reviewer for the valuable comments and suggestions. We have incorporated all the suggestions and made necessary revisions of the manuscript accordingly. Following are the point-by-point replies to the reviewer’s comments.

Reviewer 2 Report

This study evaluates three different procedures to treat vertebral compression fractures. Using finite element analyses, the stress distribution and maximum stresses are determined for vertebra osteoporotic compression fractures treated with vertebroplasty, balloon kyphoplasty, and vertebral stent. There were no significant differences among the three surgical models in stress and each treatment showed significantly lower maximum stress compared to the compressive strength of the bone cement used (PMMA). The vertebral stent displayed the highest stiffness value. However, there are significant questions about the interpretation of the results that the vertebral stent is the most effective.

General comments/questions:

  1. Figure 2 – There needs to be color code or a better description of the parts displayed. What part is the cement? What part is stent? What regions are bone and the areas where was bone removed?
  2. Section 2.1 – What was the final size of the FE model? How many nodes/elements? What was the resolution? Were the elements all the same size or was the mesh refined in any areas?
  3. Section 2.3 – Were both loading conditions done together or separately? In other words, was the vertebra compressed at 1000N and then simultaneously loaded in either flexion, extension, bending, or rotation?
  4. Figure 3 – It appears the Lateral Bending results are displayed twice, and the Extension results are missing.
  5. Only the maximum stress is reported. What is the volumetric percentage of high stress in each scenario? How many elements would be in danger of failing? Is one element with the maximum stress enough to say there may be failure? Also…
  6. What is the strength of PMMA under flexion, extension, bending, and rotation? Is the compressive strength of the PMMA enough to make a comparison to the stresses experienced under these different loading scenarios?
  7. What were the stress in the bone regions? Do the mechanical properties of the treatment influence the bone stresses?
  8. It is unclear that the data presented are enough evidence to make the statement that vertebral stent is the most effective method. Is this only because stiffness is higher? Because it is not higher by much and all treatments have significantly lower max stresses than the compressive strength of the PMMA. So it is not clear what separates the vertebral stent from the other 2 treatments.

Author Response

We would like to thank the reviewer for the valuable comments and suggestions. We have incorporated all the suggestions and made necessary revision of the manuscript accordingly. Following are the point-by-point replies to the reviewer’s comments.

Round 2

Reviewer 1 Report

The overall quality of the paper has been improved and the unclear points have been clarified. All the Reviewers observations have been addressed.

I suggest to accept the paper in the present form for publication.

Author Response

Reviewer-1:

Comments and Suggestions for Authors:

The overall quality of the paper has been improved and the unclear points have been clarified. All the Reviewers observations have been addressed.I suggest to accept the paper in the present form for publication.

Reply

Thanks for your encouraging comments. We learn a lot from your opinions and we are able to improve this article strongly.

Reviewer 2 Report

The revisions are appreciated and improve the manuscript. It would be great if the response to Comment 8 be included in the manuscript in its entirety as it clearly explains the rationale for the interpretation of the data and covers the limitations of the study.

"The interpretation of this finite element analysis can only explain immediate post-operative results but cannot predict long-term clinical effects. According to current literature review, all three surgical procedures lead to good clinical results. This is in line with the low stress distribution values found in our results. Based on clinical experience, the three procedures differ mostly by the restored vertebral height and the amount of bone cement injection. The bone cement will gradually compress over time and the height will decrease. However, metal stent covered with bone cement provides continued support to maintain vertebral height. This is the reason for superior stiffness of the VS group and may further imply long-term clinical benefits."

Author Response

Reviewer-2:

Comments and Suggestions for Authors:

The revisions are appreciated and improve the manuscript. It would be great if the response to Comment 8 be included in the manuscript in its entirety as it clearly explains the rationale for the interpretation of the data and covers the limitations of the study.

"The interpretation of this finite element analysis can only explain immediate post-operative results but cannot predict long-term clinical effects. According to current literature review, all three surgical procedures lead to good clinical results. This is in line with the low stress distribution values found in our results. Based on clinical experience, the three procedures differ mostly by the restored vertebral height and the amount of bone cement injection. The bone cement will gradually compress over time and the height will decrease. However, metal stent covered with bone cement provides continued support to maintain vertebral height. This is the reason for superior stiffness of the VS group and may further imply long-term clinical benefits."

Reply

We added this paragraph in the revised article to reflect your suggestion. (please see line 547-555)

Finally, thank you again. We feel that by taking your suggestions, we are able to improve this article strongly.